# The RUNX Family of Proteins, DNA Repair, and Cancer

**DOI:** 10.3390/cells12081106

**Published:** 2023-04-07

**Authors:** Vaidehi Krishnan

**Affiliations:** Cancer Science Institute of Singapore, National University of Singapore, Singapore 117599, Singapore; vaidehi.krishnan@duke-nus.edu.sg

**Keywords:** RUNX1, RUNX2, RUNX3, DNA repair, leukemia, Fanconi anemia, TGF-β, reactive oxygen species, p53, DNA damage

## Abstract

The RUNX family of transcription factors, including RUNX1, RUNX2, and RUNX3, are key regulators of development and can function as either tumor suppressors or oncogenes in cancer. Emerging evidence suggests that the dysregulation of RUNX genes can promote genomic instability in both leukemia and solid cancers by impairing DNA repair mechanisms. RUNX proteins control the cellular response to DNA damage by regulating the p53, Fanconi anemia, and oxidative stress repair pathways through transcriptional or non-transcriptional mechanisms. This review highlights the importance of RUNX-dependent DNA repair regulation in human cancers.

## 1. Introduction

The RUNX family of proteins comprising RUNX1, RUNX2, and RUNX3 are master regulators of development [1,2]. As transcription factors, RUNX proteins bind to the consensus (Py)G(Py)GGT(Py) sequence on DNA through the evolutionarily conserved 128-amino acid RUNT domain. CBF-beta (CBFβ or core-binding factor β) is a critical dimerization partner that allosterically enhances the DNA-binding activity of RUNX factors during transcription [3,4]. However, RUNX proteins also perform transcription-independent roles in cells; they have been shown to interact with a growing list of central developmental regulators, epigenetic enzymes, and DNA repair factors in a cell and context-dependent manner [5]. Such combinatorial protein–protein interactions allow RUNX factors to function as tunable regulators of cell growth, differentiation, and carcinogenesis [1].

Amongst the multifaceted roles of RUNX proteins in maintaining cellular homeostasis, the regulation of DNA repair is emerging as an important paradigm. RUNX proteins regulate many of the fundamental pathways regulating genomic instability, including DNA repair, oxidative stress response, replication stress response, and telomere maintenance [6,7,8]. Here, we review evidence that RUNX dysregulation actively promotes mutational accumulation in human cancers, by reducing the proficiency of DNA repair.

## 2. RUNX1 Leukemic Fusions in Hematopoietic Malignancies and Genomic Instability

RUNX1 is recurrently involved in chromosomal translocations in hematological malignancies, with almost 70 such chimeric fusions uncovered to date. The t(8;21)(q22;q22) karyotypic abnormality encoding RUNX1-RUNX1T1 (also known as RUNX1-ETO or AML-ETO) and t(12;21)(p13;q22) encoding ETV6-RUNX1 (also known as TEL-AML1) are among the most recurrent translocations in acute myeloid leukemia (AML) (10–20%) and B cell acute lymphoblastic leukemia (B-ALL) (25%), respectively [9,10]. The t(3;21) (q26;q22) translocation involving RUNX1 and MECOM1 (also known as EVI1) is frequently encountered in therapy-related myelodysplastic syndrome (MDS) and AML and during the blast crisis (BC) phase of chronic myeloid leukemia (CML) [11]. Inv16 (p13;q22), characterized by the CBFβ-MYH11 fusion, is a recurrent feature in AML (5–7%), in which RUNX function is impaired due to the inability of CBFβ to heterodimerize with RUNX [12]. Such neomorphic RUNX1 translocations mostly promote leukemogenesis by a novel gain-of-function or by dominantly inhibiting the function of the wild-type RUNX1 allele [13]. In the following section, we discuss evidence that RUNX1 leukemic fusions exacerbate genomic instability.

### 2.1. RUNX1-ETO and Genomic Instability

AML driven by RUNX1-ETO or AML1-ETO is a very well-studied AML subtype [14]. The RUNX1-ETO fusion gene has a structure comprising of the RUNX1 DNA-binding domain in addition to four conserved domains of the ETO protein, termed NHR1 to NHR4, that recruit transcriptional repressor complexes such as NCOR/HDAC/mSIN3a [15]. Since the RUNX1-ETO protein retains the DNA-binding domain of RUNX1 but lacks the carboxyl-terminal transactivation domain, the fusion binds to several RUNX1 target genes but represses their expression [16,17,18,19] and functions as a regulator of self-renewal and differentiation. 

Notably, amongst the various RUNX1 leukemogenic fusions, the clearest mechanistic links between RUNX dysfunction and a “mutator” phenotype exist for t8;21 AML (Figure 1) [20]. However, RUNX1-ETO expression requires additional co-operating mutations in *KIT, FLT3, RAS, ASXL1*, and *ZBTB7A*, -9q, or –Y for the complete leukemic transformation of cells, and consistent with this idea, additional chromosomal aberrations are detected in almost 70% of t(8;21)-positive AML [21]. 

Several studies have convincingly shown that RUNX1-ETO drives the acquisition of such co-operating mutations by downregulating the fidelity of DNA repair, thereby promoting a “mutator phenotype”. For instance, the overexpression of RUNX1-ETO reduced the expression of 17 DNA repair genes that participate in several DNA repair pathways, of which eight genes were involved in base excision repair (BER) (*ADPRTL2*, *FEN1*, *OGG1*, *MPG*, *LIG3*, *POLB*, *POLD2*, and *POLD3*) [22]. In assays that directly measure DNA repair, RUNX1-ETO-expressing cells were impaired in repairing oxidative lesions and had elevated levels of γH2AX, a marker of DNA double-strand breaks (DSBs) [22]. In an independent study by Krejci et al., RUNX1-ETO expression reduced the expression of genes from the ATM, ATR, and Fanconi anemia (FA) pathways of DNA repair [23]. Interestingly, HLTF, a protein that promotes replication fork reversal and limits multiple mechanisms of unrestrained DNA synthesis and replication stress, was also identified as a target of RUNX1 and was downregulated by RUNX1-ETO [24,25]. Likewise, Esposito et al. reported the downregulation of multiple FA and homologous recombination (HR) genes, including *RAD51 and BRCA1/2*, and the DSB sensor *ATM*, in RUNX-1 ETO-expressing leukemia [26]. 

In subsequent studies, to directly quantify the rate at which mutations are acquired, Forster et al. expressed RUNX1-ETO in the non-transformed TK6 lymphoblastoid cell line, and mutations at the *PIG-A* reporter gene was used as the read-out for genomic instability [27]. Remarkably, RUNX1-ETO expression was sufficient to predispose cells to elevated mutational acquisition both spontaneously and after exposure to genotoxic agents [27]. Likewise, in an in vivo model, RUNX1-ETO overexpression in a LacZ-plasmid (pUR288) expressing transgenic mouse resulted in an approximately 2-fold higher mutation rate over controls’ [23]. Independently, reactive oxygen species (ROS) have also emerged as a major etiology driving mutational accumulation in pediatric AML driven by RUNX1-ETO. Whole genome sequencing revealed that RUNX1-ETO-positive cases were associated with a higher prevalence of the ROS-associated SBS18 mutational signature, specifically owing to a high frequency of C>A transversions in this AML subtype. Additionally, it was shown that within RUNX1-ETO-fusion-positive AML cases, ROS-associated processes were not only contributing to mutations but also pro-oncogenic effects [28,29]. Together, the above studies highlight how the RUNX1-ETO oncoprotein exacerbates genomic instability through not one but multiple mechanisms, thereby permitting preleukemic cancer cells to acquire secondary hits which promote malignant progression. 

### 2.2. ETV6-RUNX1 and Genomic Instability

The chimeric fusion protein ETV6-RUNX1 is a hallmark of B-ALL in which the N-terminus of the ETV6 gene is fused to almost the entire RUNX1 protein, and this event is thought to convert RUNX1 from a transcriptional activator to a repressor [30]. The early co-operating processes involved in the pathogenesis of this fusion protein have been difficult to determine. However, recent lineage tracing studies in mice have shown that the ETV6-RUNX1 clone is mostly preleukemic, and a second oncogenic hit appears essential for leukemic transformation [31].

A ETV6-RUNX1transgenic mouse model in which the fusion protein was expressed in precursor CD19+ B cells was examined for evidence of DNA damage accumulation and genomic instability [32]. Consistently, higher ROS and DNA damage accumulation were evident in the ETV6-RUNX1overexpressing mice, supporting the concept of higher mutability of the ETV6-RUNX1expressing genome (Figure 1). Independently, microarray comparisons between *ETV6-RUNX1* knockdown and control ALL lines also revealed “DNA damage response” as a significant term. In this study, a total of 777 genes were substantially altered upon ETV6-RUNX1knockdown, and these comprised the DNA damage response genes (*DRAM1*, *MDM2*, *CDKN1A*, and *PSD4*) and genes regulated by p53, such as *DDIT4* [33]. As further corroborating evidence, p53 signaling emerged as one of the central pathways deregulated in ETV6-RUNX1 expressing B-ALL compared to the fusion-negative B-ALL counterparts. Specifically, ETV6-RUNX1upregulated the transcription of *MDM2*, the negative regulator of p53; consistently blocking MDM2 through the inhibitor, nutlin, caused a surge in apoptosis [34]. Thus, ETV6-RUNX1-dependent p53 signaling impairment appears to be one of the driving forces underlying the development of a second oncogenic hit in this class of leukemia. In future work, a transgenic mouse model for ETV6-RUNX1generated in a p53-deficient background can clarify the precise contribution of p53 to ETV6-RUNX1-driven leukemogenesis. 

### 2.3. RUNX1-EVI1 and Genomic Instability

The expression of RUNX1-EVI1 is common in therapy-induced MDS and during the BC transformation of chronic phase (CP) CML [35,36]. In this fusion protein, the N-terminal RUNT domain of RUNX1 is fused to almost the entire EVI1. In recent studies by Kellaway et al., RUNX1-EVI1 binding was shown to cause a redistribution of wild-type RUNX1 binding, which interfered with both the *RUNX1* and *EVI1* transcriptional programs [37]. While gene expression changes in DNA damage response genes were not reported in this study, a microarray analysis comparison of RUNX1-EVI1-driven transcriptional changes in a zebrafish model revealed the altered expression of mismatch repair (MMR) and nucleotide excision repair (NER) genes [38] (Figure 1). Moreover, given that RUNX1-EVI1 translocation is frequently retrieved after conventional chemotherapy, such as following hydroxyurea treatment in CML, it is tempting to speculate that this fusion gene provides a competitive advantage in the presence of DNA damage. Consistently, a proteomic analysis of the EVI1 protein-binding complexes unveiled EVI1 interaction with components of DNA repair and recombination [39]. One can speculate that RUNX1-EVI1 might create genomic instability in cancers by altering the interaction of EVI1 with DNA repair factors, although this model requires validation through rigorous biochemical studies.

## 3. RUNX1 Mutations in Cancers and Genomic Instability

In addition to translocations, sporadic somatic mutations in RUNX1, either monoallelic or biallelic, are found in multiple leukemias, including AML (6–33%) [40]. Germline monoallelic RUNX1 mutations, on the other hand, strongly predispose families to familial platelet disorder with associated acute myeloid leukemia (FPD/AML). A third class of mutations is secondary therapy-induced, and these are most commonly observed in MDS. Somatic RUNX1 mutations have also been identified in solid tumors of the breast (4%), esophageal (7%) endometrial, and ovarian cancers and contribute to both drug resistance and disease progression. Given that sporadic RUNX1 mutations are rarely initiating events in leukemia, and *RUNX1* deficiency often requires secondary hits such as MLL-ENL, NRAS, and EVI5 mutations [41,42,43], it has been postulated that RUNX1 mutations may trigger genomic instability in human cancers, which in turn, renders such preleukemic cells permissive for the accumulation of tumor-promoting secondary hits.

RUNX1 mutations can be missense, frameshift deletions, insertion, nonsense, or splicing mutations, and the consequence of some of these can be predicted based on the known domains of RUNX1 [40,44]. For example, missense mutations that cluster around the RUNT domain (R174Q mutation), impair DNA binding and transcription. Similarly, hypomorphic mutations at the RUNX–CBFβ binding interface (S67I, S67R mutations) reduce the affinity of RUNX binding to DNA and, thus, attenuate transcription. However, RUNX1 mutations are also found at the C-terminus, and rarely some mutations may even result in a longer protein due to frameshift insertions that disrupt the stop codon; the effects of such events are mostly unclear [44].

### 3.1. RUNX1 C-Terminal Deletions and Genomic Instability

To specifically study the consequence of C-terminal RUNX1 deletions in cancers, Satoh et al. expressed a deletion mutant lacking 225 amino acids (RUNX1dc) from the C-terminus of RUNX1 in murine hematopoietic stem and progenitor cells (HSPCs) [45] (Figure 2). Notably, gene expression profiling revealed that *RUNX1dc* expression resulted in γH2AX accumulation and the suppression of *GADD45a*, a p53-responsive gene and regulator of the NER pathway. Moreover, after ultraviolet (UV) light exposure, RUNX1dc-expressing cells accumulated elevated levels of (6–4) photoproducts (6–4 PPs) and cyclobutane pyrimidine dimers, which are major products of DNA damage induced by UV-B. The clonogenic ability of cells expressing RUNX1dc was significantly reduced after exposure to a cross-linking agent, cisplatin. These findings led the authors to conclude that RUNX1dc, which was originally found in MDS patients, attenuated the fidelity of DNA repair and promoted AML.

### 3.2. RUNX1 RUNT Domain Mutations and Genomic Instability

Antony-Debre et al. examined the effects of a dominant negative RUNT domain mutation (RUNX1-R174Q) in an induced pluripotent stem cell (iPSC) model. Here, upon RUNX1-R174Q expression, genes related to p53-dependent gene expression decreased, while genomic instability within the granulomonocytic cell population increased [46] (Figure 2). Interestingly, the RUNX1-R174Q overexpression phenotype in this iPSC cell line phenocopied the downregulation of wild-type *RUNX1* in an H9 ESC cell line and an FPD/AML iPSC cell line with monoallelic RUNX1 deletion. Thus, it was proposed that the lower residual activity of the wild-type RUNX1 protein resulted in genomic instability and impacted the risk for leukemia development in FPD/AML patients. 

### 3.3. RUNX1 CML Blast Crisis Mutations and Genomic Instability 

Awad et al. studied the role of three RUNX1 missense mutations (p.R162K, p.R204Q, and p.R107C) and one nonsense mutation (p.K117 *), all of which were located within the RUNT domain in BC-CML [47]. A comparison between mutant *RUNX1* and wild-type *RUNX1* samples revealed the upregulation of a mutation signature 9 in RUNX1 mutant samples (Figure 2). Signature 9 is responsible for somatic hypermutation caused by polymerase η and AID/RAG activity. Thus, genomic instability observed in BC-CML can at least partly be attributed to RUNX1 mutations.

### 3.4. RUNX1 MDS Mutations and Genomic Instability

Recently, to identify novel biomarkers of MDS progression, Kaisrlikova et al. conducted a comprehensive comparison between MDS patients with a lower or higher risk for progression [48]. Notably, RUNX1 mutations were identified as the major predictor of rapid progression. RUNX1-unmutated MDS patients were protected by DNA damage and cellular senescence, which emerged as a critical anticancer barrier to cancer progression. Thus, in the context of MDS, RUNX1 mutations contributed to malignant transformation by interfering with an anticancer barrier.

## 4. RUNX3 Defects in Human Cancers and Genomic Instability

Unlike RUNX1, which is frequently mutated in human cancer, the *RUNX3* gene is often transcriptionally silenced in cancer by CpG island DNA methylation or through EZH2-dependent H3K27me3 (histone H3 lysine 27 trimethylation) modification in the RUNX3 promoter [49]. RUNX3 can also be inactivated by cytoplasmic mis-localization and rarely through mutational inactivation (R122C mutation), as evident in gastric cancer [50,51]. DNA-damaging assaults such as smoking and ROS were shown to induce RUNX3 promoter hypermethylation [52,53]. Recently, in a novel mechanism of *RUNX3* inactivation, Lee et al. showed hypoxia-induced methylation of RUNX3 by the enzyme G9a, and the methylated RUNX3, in turn, was attenuated in transactivation [54,55]. In this study, RUNX3 protein methylation was correlated with increased proliferation and initiation of tumorigenesis. Interestingly, *Helicobacter pylori* (*H. pylori*) infection itself has been shown to trigger RUNX3 inactivation through gradual step-wise promoter hypermethylation and accompanying silencing of the gene. Since *H. pylori* infection downregulates the expression of several DNA repair genes [56,57], it can be speculated that some of these transcriptional changes may be related to *RUNX3* silencing, a model that needs to be experimentally tested in future work. Overall, based on the above observations, it can be hypothesized that *RUNX3* silencing upon DNA damage might relieve a key anticancer barrier in epithelial tissues. Consistently, RUNX3 methylation was proposed as a ‘clock’ to determine the rate of bladder cancer progression [58].

Intriguingly, RUNX3 functions as an oncogene in NKT cell lymphoma, osteosarcoma, and ovarian cancers, mainly by increasing the transcription of *MYC*. By creating a transgenic mouse model, Douchi et al. showed that mutant RUNX3 R122C protein promotes gastric hyperplasia, and the upregulation of *MYC* was noted [51]. Thus, *MYC* upregulation is a common theme that is emerging when the downstream consequences of oncogenic RUNX3 expression are being examined. Given that MYC activation is associated with DNA replication stress and DNA damage [59], RUNX3 might activate DNA damage in these models through the transcriptional regulation of *MYC* (Figure 3).

### 4.1. RUNX3 Inactivation and Genomic Instability

To examine how *RUNX3* expression levels are related to genomic instability in human cancer, Tay et al. conducted a comprehensive genome-wide analysis of TCGA (The Cancer Genome Atlas) datasets [60]. Correlation coefficients between *RUNX3* transcript levels and copy number alterations (CNA) or mutation counts were computed for all cancers. This analysis revealed that CNAs negatively correlated with *RUNX3* expression most significantly in bladder urothelial carcinoma (BLCA) (*n* = 404, *p* = 9.28 × 10^−6^) and lung adenocarcinomas (LUAD) (*n* = 513, *p* = 8.78 × 10^−7^) (Figure 3). On the other hand, mutation rate negatively correlated with *RUNX3* expression most significantly in esophageal carcinoma (ESCA) (*n* = 184, *p* = 4.3 × 10^−6^) and liver hepatocellular carcinoma (LIHC) (*n* = 366, *p* = 8.4 × 10^−5^). It can be argued that RUNX3 suppresses genomic instability more often in cancers having an etiological link to DNA damage, such as lung, bladder, and esophageal cancers which are predisposed by smoking, alcohol, and interstrand crosslinking agents, respectively. It can be hypothesized that in such cancers, lower RUNX3 probably lowers proapoptotic signaling by p53, allowing cells to survive in the presence of DNA damage (see below, RUNX and p53 crosstalk). 

### 4.2. RUNX3 Activation and Genomic Instability 

In contrast, RUNX3 was identified as a super-enhancer-associated oncogene in AML and was one of the most highly expressed genes in this cancer type. As a transcription factor, RUNX3 is bound to the promoter of cell cycle-related genes in both normal and AML cells. However, within AML cells, RUNX3 is also bound to the promoters of DNA repair genes (*CHEK1*, *RAD51C*, *RPA2*, and *DDB1*), antiapoptotic genes (*BCL2*, *BCL2L10*, *BCL2L12*, and *MCL1*), and genes implicated in leukemogenesis (*MYC*, *CD93*, *KIT*, *IKZF2*, *FTO*, and *SOX4*) [61] (Figure 3). In this study, *RUNX3* knockdown inhibited leukemic progression by inducing DNA damage and higher apoptosis. Thus, as an oncogene, *RUNX3* overexpression induces a higher resistance to DNA damage-induced apoptosis. It can be inferred from the above studies that *RUNX3* levels are critical in determining whether DNA damage signals enter the apoptotic pathway via p53. The influence of RUNX proteins on p53 signaling strengths will be discussed in the following sections.

## 5. RUNX2 Defects in Cancers and Genomic Instability

In contrast to RUNX1 and RUNX3, which may function as oncogenes or tumor suppressors, RUNX2 is mostly overexpressed and oncogenic in human cancer. RUNX2 has promigratory effects on breast, prostate, and thyroid cancer cells, osteosarcoma, and melanoma cells and has emerged as a key regulator of cancer metastasis. *RUNX2* overexpression increases the expression of genes involved in invasion and metastasis, such as *MMP9*, *MMP13*, *OPN*, *VEGF*, and *IL-8*, epithelial-mesenchymal transition factors such as *SNAI2*, *SMAD3*, and *SOX9*, and motility genes such as *FAK/PTK2* and *TNL1*. RUNX2 also promotes metastasis by activating the AKT/PI3K, YAP-TAZ, TGFβ, and WNT signaling pathways and angiogenesis, thereby driving positive feedback loops that advance cancer progression [62].

### RUNX2 Dysregulation and Genomic Instability

In one of the earliest studies on the relationship between RUNX proteins and DNA repair, primary RUNX2-null osteoblasts were shown to accumulate spontaneous γH2AX foci, experience loss of telomere integrity, and have delayed DNA damage response [63] (Figure 3). Subsequently, RUNX2 was shown to form functional complexes with BAZ1B, RUVBL2, and INTS3 and influences UV repair by complexing with H2AX and decreasing histone H3 lysine 9 acetylation levels [64]. More recently, RUNX2 was shown to promote the phosphorylation of H2AX at 142, thus favoring apoptosis instead of repair [65]. In this study, RUNX2 recruitment to osteogenic target genes was dependent on DNA damage and led to an enhancement in calcification during aging and chronic disease. In the context of malignancy, overexpressed *RUNX2* regulates chemosensitivity by attenuating the transcriptional activity and proapoptotic function of p73 after exposure to the chemotherapeutic adriamycin, supporting an oncogenic role for RUNX2 in chemoresistance [66]. However, it remains unknown if *RUNX2* overexpression is causally relatedly to genomic instability and mutational accumulation in human cancers. 

## 6. Molecular Mechanisms Underlying RUNX Dysregulation and Genomic Instability in Cancer

RUNX proteins regulate the cellular response to DNA damage by participating in the p53, Fanconi anemia, and oxidative stress repair pathways both by canonical transcriptional regulation and in a noncanonical manner by engaging in novel protein–protein interactions. In the following sections, the mechanisms by which RUNX proteins promote the proficiency of DNA repair are summarized.

### 6.1. RUNX and p53-A Crosstalk between Two Tumor Suppressors

RUNX factors were investigated as regulators of p53-dependent DNA damage response in several studies [67]. As well-known, p53 can transcriptionally regulate the expression of multiple cell cycle and apoptosis genes after DNA damage which are needed for the maintenance of genomic integrity. RUNX1 was recruited with p53 to p53-dependent target gene promoters and co-operatively promoted the transactivation of p53 target genes *BAX*, *PUMA*, *NOXA*, and *p21* (Figure 4) [68]. The loss of *RUNX1* attenuated p53 acetylation at Lys-373/382 by p300 and reduced doxorubicin-dependent apoptosis. Satoh et al. showed that RUNX1 and p53 synergistically activate *Gadd45a*, a sensor of DNA damage [45]. In contrast to the tumor suppressor role for RUNX1 in the above studies, Morita et al. showed that the RUNX1-p53-CBFβ regulatory loop was oncogenic in AML. p53 and CBFβ are upregulated in response to RUNX1 depletion, and their mutual interaction causes physiological resistance against chemotherapy for AML [69]. Similar to RUNX1, RUNX3 was also found to regulate p53-mediated responses following exposure to doxorubicin [70]. Here, *RUNX3* knockdown inhibited DNA damage-dependent apoptosis in p53 wild-type cells but not in p53-deficient cells. RUNX3 and p53 were found to co-immunoprecipitate as a complex, and RUNX3 induced the phosphorylation of p53 at Ser-15, promoting p53-dependent apoptosis (Figure 4). The close interactions between RUNX1/2 with p53 and the implications for DNA damage and cancer progression have been described in detail recently [71]. In contrast to RUNX1 and RUNX3, the complex between RUNX2 and p53 repressed the transcription of p53-inducible genes, such as *p21 WAF1* and *BAX*, in response to adriamycin via the recruitment of HDAC6 [72]. Thus, the knockdown of *RUNX2* significantly enhanced adriamycin-mediated apoptotic cell death in U2OS cells, a phenomenon that was contradictory to the phenotypes exhibited upon the knockdown of *RUNX1* and *RUNX3*.

### 6.2. RUNX Proteins and the Fanconi Anemia Pathway of DNA Repair

During the genetic analysis of mice doubly-deficient for *Runx1* and *Runx3*, an unexpected role for RUNX proteins in the regulation of the Fanconi anemia (FA) pathway of DNA repair was discovered (Figure 5). The FA pathway of DNA repair comprises a core set of proteins that mediate the recognition and resolution of DNA interstrand crosslinks (ICLs). While RUNX proteins have not been found mutated in human FA, at least two rare cases of FA were presented with deletions in the genomic region encompassing the *RUNX1* gene [73,74]. FA patients manifest bone marrow failure (BMF), MDS, and AML, while cells isolated from such patients show an acute sensitivity to DNA ICLs [75]. Very similar to FA patients, mice that were a double-knockout (DKO) for *Runx1* and *Runx3* genes experienced mortality due to either BMF or a myeloproliferative disease, and cells derived from such mice had an elevated sensitivity to DNA ICLs [76]. At the molecular level, RUNX1/3 DKO cells were deficient in the recruitment of monoubiquitinated FANCD2 (Ub-FANCD2) to sites of DNA repair, a central step essential for the successful repair of ICLs. Since RUNX proteins associate with FANCD2 in a DNA damage-dependent manner at the chromatin, importantly forming a complex *independent* of CBFβ, the authors proposed a nontranscriptional role for RUNX in the FA repair pathway. 

As further direct biochemical evidence, Tay et al. assembled DNA structures that resembled intermediates of ICL repair, such as “splayed-arm” DNA, using DNA sequences that lack the consensus (Py)G(Py)GGT(Py) site that RUNX proteins bind to during transcription [60]. Intriguingly, RUNX proteins gained the ability to bind to splayed arm DNA in vitro upon exposure to DNA ICLs in a PARP-dependent manner. The authors found that RUNX proteins were poly(ADP) ribosylated or PARylated after DNA damage (Figure 5), and the Aspartate 103 (RUNX1D103) residue was at least one of the critical sites PARylated by PARP-1. It was proposed that the RUNX PARylation modification was catalyzed at the site of ICLs, and PARylated RUNX, in turn, recruited Ub-FANCD2 to sites of DNA damage. In an independent study, RUNX2 was also found to be PARylated in response to DNA damage, and PARylated RUNX2 regulated osteogenic gene expression contributing to age-related osteogenic pathologies [65]

The RUNX-FA crosstalk has several clinical implications in the regulation of genomic integrity in epithelial and hematological cancers, as listed below. First, in breast cancers, RUNX1 walker domain mutations (G141 and R142 residues) were defective for DNA damage-dependent RUNX PARylation [60] (Figure 5). Such cancers may be sensitive to ICL agents. Second, RUNX1-ETO expression impaired FANCD2 recruitment after DNA damage [76], decreased the expression of FA/HR genes, and elicited PARP inhibitor sensitivity [26]. Third, in FA patients progressing from MDS to leukemia, cryptic *RUNX1* lesions (translocations, deletions, or mutations) were observed [77,78]. It is possible that RUNX1 aberrations co-operatively increase genomic instability in FA patients and allow step-wise clonal selection of leukemic cells. Lastly, besides ICL repair, FANCD2 is needed for replication fork protection, in parallel with the BRCA2 pathway to stabilize the Rad51 nucleoprotein filament [79]. It remains to be tested if the loss of RUNX proteins attenuates replication fork protection mediated by FANCD2 and RAD51, especially in BRCA1/2-deficient breast cancers. Taken together, RUNX dysfunction can be an alternative route by which the FA pathway is inactivated in cancers, and an in-depth analysis of RUNX PARylation and its role in DNA repair may lead to new therapeutic opportunities.

### 6.3. The Association of RUNX Proteins with DNA Repair Complexes

An orthogonal biochemical approach provided additional evidence on the potential roles of RUNX proteins in DNA repair [60]. It was interesting that RUNX3 interactome studies unveiled their physical association with several DNA repair complexes both in the absence and presence of DNA damage. RUNX3 was expressed in an inducible manner, and RUNX3-co-immunoprecipitates were analyzed in the absence or in the presence of DNA ICLs. SILAC (stable isotope labeling by amino acids in cell culture)-based mass spectrometry methodology was used to analyze RUNX3-immunoprecipitate fractions. Even in the absence of DNA damage stimulation and under basal conditions, RUNX3 interacted with several proteins involved in DNA repair and DNA replication, amongst others, with “DNA repair” emerging as the top enriched term in a “pathway analysis” of RUNX3-interacting proteins (Figure 6).

RUNX3 also bound to components of the nonhomologous end joining (NHEJ) and alt-end joining pathways, such as PRKDC (DNA-PKcs), XRCC5 (KU70), XRCC6 (KU80), and LIG3, of which the interaction with RUNX3 and KU70 was consistent with earlier studies [81]. Of note, the interaction of RUNX proteins with NHEJ factors was DNA-independent given that the co-immunoprecipitations were performed in the presence of the nuclease, benzonase, which is likely to have disrupted any interactions that may be retrieved indirectly due to DNA binding. RUNX proteins presumably associate with NHEJ factors in the nucleoplasm, and this complex may have novel functions in DNA repair. It is pertinent to note that DNA-PKcs (PRKDC), for instance, have additional roles in DNA damage-induced autophagy, cell cycle progression, and mitosis [82,83]. RUNX3 also interacted with the helicases DDX17, DDX5, and DDX20, of which DDX5 has been shown to resolve R loops at DSBs to promote DNA repair and prevent chromosomal deletions [84] (Figure 6). Similarly, the interaction of RUNX proteins with the RFC subunits RFC3 and RFC5, with the condensins SMC2, SMC3, and SMC1A, and with histones suggest broad roles of RUNX proteins in DNA replication, chromosome condensation, and chromatin modeling, many of which have fundamental roles in the maintenance of genomic integrity. Further biochemical studies are needed to biochemically dissect the RUNX3 interactome in the context of DNA repair and examine their coregulation with RUNX-driven transcription.

Intriguingly, in the presence of DNA damage, RUNX3 has an increased interaction with the kinetochore complex (AURB, DSN1, CASC5, MIS12, ZWINT, PMF1, NSL1, and TPX2), the E2F7 transcription factor, and BLM (Figure 5) [60]. Given that the kinetochore complex is a regulator of the spindle checkpoint [85], it can be hypothesized that RUNX proteins may prevent chromosome mis-segregation in the presence of DNA damage by activating the spindle checkpoint. Likewise, the novel complex between RUNX3 and E2F7 might modulate cell-cycle progression since E2F7 is a transcriptional repressor that blocks cell-cycle progression in conjunction with p53 [86]. The higher interaction between RUNX proteins and BLM was studied in greater detail, and these two proteins were shown to regulate FANCD2 recruitment in an epistatic manner [60,87]. In a recent study on Primpol, a primase-polymerase required for DNA damage tolerance, a novel nuclear complex containing Primpol, RUNX1, BLM, RMI1, and RMI2 was identified [88]. Notably, the interaction between Primpol and RUNX1 increased after DNA damage, while the interaction between Primpol and BLM decreased after DNA damage.

Moving forward, a temporal dissection of the RUNX-interactome after DNA damage will likely shed a novel mechanism by which RUNX proteins fine-tune the fidelity of the FA/HR pathway at distinct steps. At the early stages of DNA repair, a transient association between Ub-FANCD2 with RUNX/BLM complex might regulate FANCD2 recruitment to sites of DNA damage. On the other hand, at the later stages of repair, RUNX proteins may have additional roles in the regulation of cell cycle progression and transcription.

### 6.4. TGFβ Driven Epithelial-Mesenchymal Transition (EMT), RUNX1/RUNX3 Loss, and DNA Damage

TGF*β* is a cytokine that is released by the stromal cells from the tumor microenvironment and is a promoter of cancer progression, EMT, and drug resistance in advanced cancers. To investigate the tumor suppressor function of RUNX proteins in advanced lung cancers, Krishnan et al. used a TGF*β*-inducible EMT as the cellular model [89,90]. Unexpectedly, *RUNX1* and *RUNX3* inactivation was sufficient to trigger dramatic DNA damage and genomic instability, specifically when cells were exposed to TGFβ and underwent EMT (Figure 7). In mechanistic studies, it was found that TGF*β* induces oxidative stress, which is usually quenched to lower levels by the antioxidant *HMOX1*, a transcriptional target of RUNX proteins. However, when lung cancer cells lacked RUNX3 proteins, TGF*β* induced much higher levels of oxidative damage, which gets converted into DNA DSBs upon collision with replication forks. Thus, RUNX proteins function as tumor suppressors by limiting the oxidative stress induced by TGF*β*.

To understand the clinical implications of the above findings, a genomic analysis of TCGA datasets was conducted. It was found that lung cancers with a TGF*β* signature were more likely to experience genomic instability and mutational accumulation when they had lower expression levels of *RUNX3*. Notably, in this model, DNA damage triggered cellular senescence, which led to tumor-promoting inflammatory cytokine expression and acquisition of the senescence-associated secretory phenotype (SASP), represented by *CXCL1*, *INHBA*, *BMP2*, *CCL2*, *CXCL3*, *CXCL2*, *IL32*, *IL8*, *AREG*, *GDF15*, and *IL1A* upregulation. Since inflammation can induce DNA damage, RUNX3 deficiency creates a self-re-enforcing feedback loop of inflammation and genomic instability.

### 6.5. Oxidative Stress and RUNX Factors

The role of RUNX factors in oxidative stress response has been examined under diverse cellular contexts. During the transcriptional analysis of RUNX1-ETO targets, it was found that this chimeric fusion suppresses the repair of oxidative DNA damage by repressing BER genes such as *OGG1* [22]. *OGG1* is an 8-oxoguanine DNA glycosylase that is required in the BER pathway to remove oxidized guanine nucleotides formed under oxidative stress. *OGG* downregulation was also evident in t8;21 patients [91]. In foreskin fibroblasts (Hs68), the overexpression of *RUNX1* resulted in heightened oxidative stress, p38 MPAK activation, and premature senescence [92]. In contrast, in breast epithelial acinar morphogenesis, *RUNX1* and *FOXO1* inhibition synergized to cause heightened oxidative stress during 3D morphogenesis in vitro [93]. Interestingly, RNT-1, the *C.elegans* RUNX homolog, is stabilized by oxidative stress through the MAPK pathway [94]. It was proposed that RNT-1 stabilization facilitated rapid response to environmental stress challenges in the intestine. Taken together, the above results imply that the relationship between RUNX and oxidative stress is tissue and context-dependent, and more studies are needed to fully dissect the role of RUNX proteins in the oxidative stress response pathway.

### 6.6. RUNX Deficiency, Inflammation, and DNA Damage

Given that inflammation is a strong inducer of oxidative stress and DNA damage, the loss of RUNX proteins can indirectly induce genomic instability by elevating the levels of inflammation. Consistently, several reports link RUNX dysregulation with inflammation and genomic instability. In a mouse model of RUNX1-ETV6, an inflamed niche (IL6/TNFα/ILβ) was implicated in generating DNA damage and in predisposing preleukemic clones to leukemic transformation [95]. Independently, Fitch et al. showed that IL-10 deficiency increased the expression of proinflammatory cytokines, which caused DNA damage, mutational accumulation, and B cell neoplasms in ETV6-RUNX1 mice [96]. Using mouse models, it has been shown that two transcriptional enhancers of the genes *CCL4* and *CCL5* were negatively regulated by the RUNX/CBFβ transcription factor complexes [97]. Given that the CCR5/CCL5 axis regulates DNA damage repair and breast cancer stem cell expansion [98], it remains to be tested whether RUNX dysregulation can elicit DNA damage in cancer stem cells through the transcriptional regulation of *CCL5*.

## 7. Conclusions and Perspectives

In conclusion, this article highlights the critical role of the RUNX family of transcription factors in maintaining genomic integrity in human cancer (summarized in Table 1). The four distinct modes of action of RUNX proteins in DNA repair demonstrate their versatility in preventing genomic instability. First, RUNX proteins transcriptionally regulate the expression of genes involved in DNA repair, including p53 targets. Second, RUNX proteins undergo DNA damage-dependent PARylation by PARP1 and participate in the Fanconi anemia pathway of DNA repair. Third, RUNX proteins establish interactions with DNA repair complexes in unstressed cells, implying potential roles in genomic maintenance even in the absence of extrinsic DNA damage. Lastly, RUNX proteins suppress inflammation, which is a strong source of oxidative stress and DNA damage. Consistent with these functional roles in DNA repair, the loss of RUNX factors can trigger genomic instability in various epithelial cancers and leukemia, increasing the risk of cancer development. Studying alternative DNA repair pathways that are necessary for the survival of RUNX-dysregulated cells may provide new opportunities for targeted therapies against such cancers.

## Figures and Tables

**Figure 1 cells-12-01106-f001:**
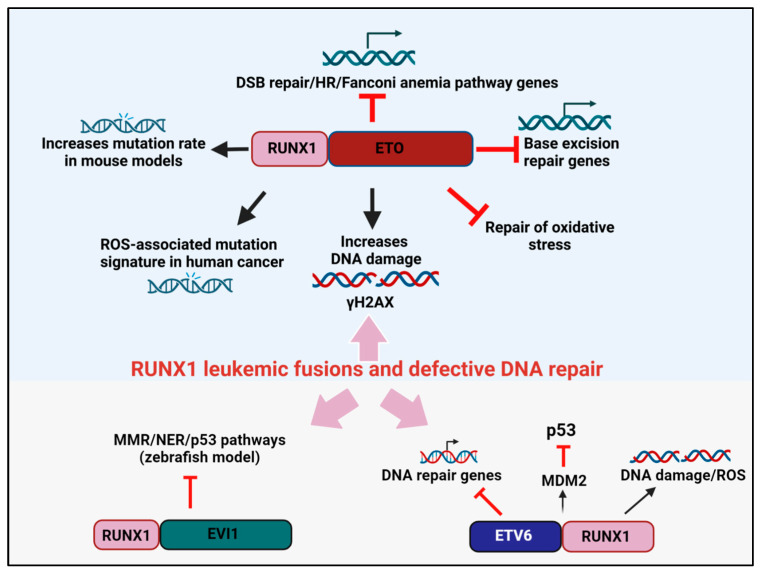
RUNX1 leukemic fusion proteins, DNA repair, and cancer. RUNX1-ETO attenuates the expression of genes involved in base excision repair, DSB repair, and the HR/FA pathways. RUNX1-ETO overexpression also reduces the efficiency of oxidative stress repair and induces the accumulation of γH2AX marked DSBs. In mouse models, RUNX1-ETO expression elevates mutation rates, while in human cancers, RUNX1-ETO expression induces a ROS-associated mutation signature. The leukemogenic fusion protein ETV6-RUNX1 blocks the expression of DNA repair genes and attenuates p53 signaling by increasing *MDM2* expression. RUNX1-EVI1 was shown to reduce the expression of genes involved in MMR and NER in a zebrafish model.

**Figure 2 cells-12-01106-f002:**
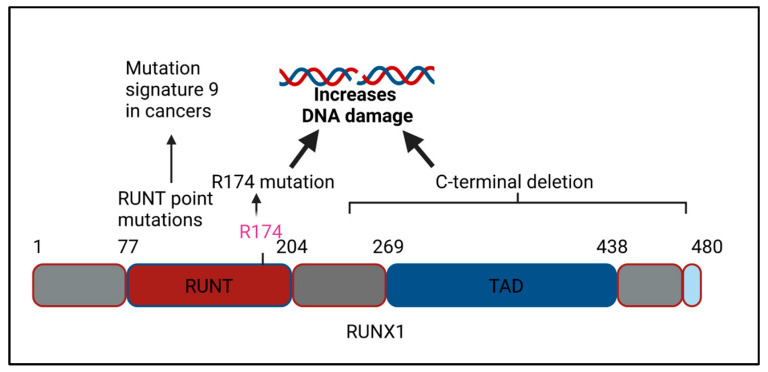
RUNX1 deletions, mutations, DNA repair, and cancer. A C-terminal deletion mutant of RUNX1 (RUNX1dc), as well as the DNA-binding point mutant of RUNX1 (R174Q), has been shown to increase DNA DSBs in the cells. Moreover, RUNT domain point mutations in blast crisis CML induce the mutational signature 9, which is attributed to polymerase η and AID activity during somatic hypermutation (SHM).

**Figure 3 cells-12-01106-f003:**
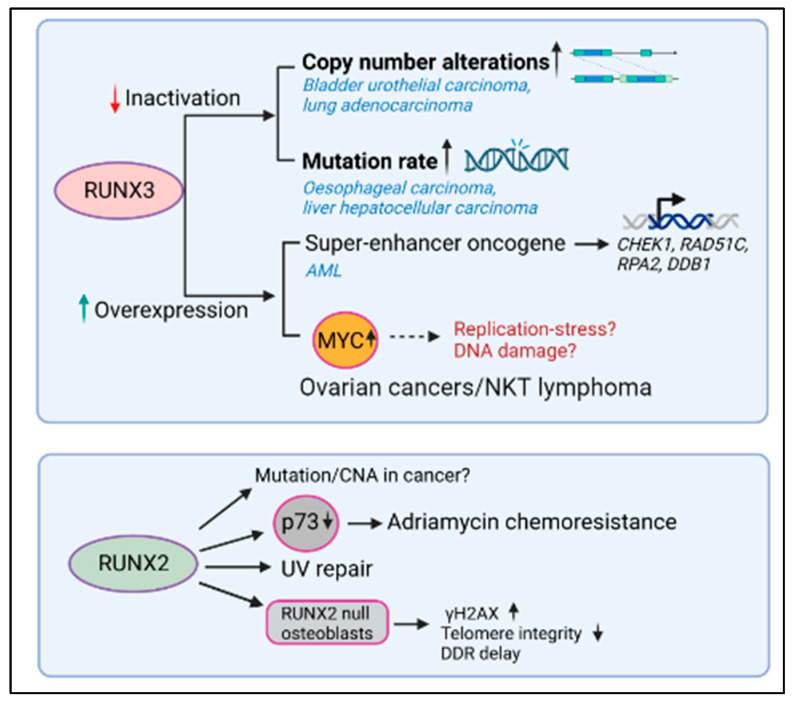
RUNX3 and RUNX3 dysregulation, DNA repair, and cancer. Lower *RUNX3* transcript levels correlated with higher copy number alterations (CNAs) in bladder, urothelial carcinoma, and lung adenocarcinoma and with high mutation rate in esophageal carcinoma and liver hepatocellular carcinoma. In contrast, the higher levels of *RUNX3* correlated with reduced DNA repair gene expression in AML and with higher levels of *MYC* in ovarian cancer and NKT cell lymphoma. While the expression of *RUNX2* promoted the repair of UV-induced DNA damage and induced greater chemoresistance to adriamycin, the relationship between RUNX2 levels and mutational accumulation in the context of human malignancy remains unknown.

**Figure 4 cells-12-01106-f004:**
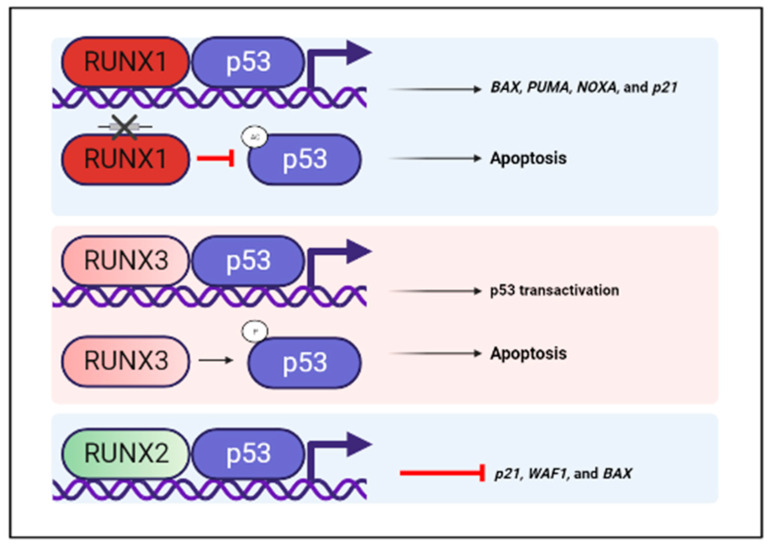
RUNX-p53 crosstalk and regulation of the DNA damage response. Both RUNX1 and RUNX3 form a complex with p53 and promote the transactivation of p53 target genes upon exposure to DNA damage. In addition, RUNX1 stimulates the acetylation of p53 at Lys-373/382, while RUNX3 promotes p53 phosphorylation at the serine 15 residue. The interaction of RUNX2 with p53, on the other hand, suppresses the transactivation of p53 target genes in response to the chemotherapeutic adriamycin.

**Figure 5 cells-12-01106-f005:**
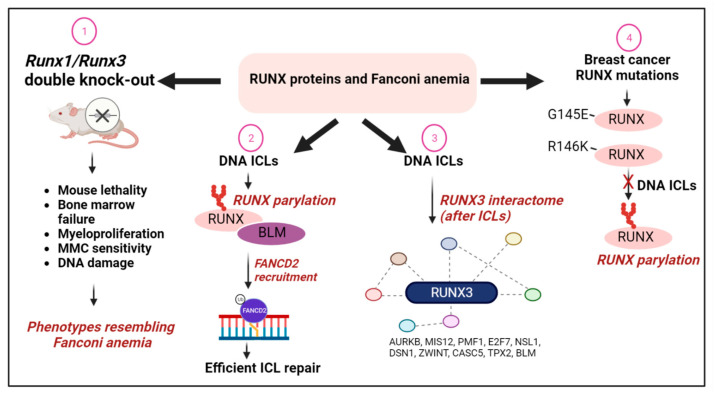
RUNX proteins and the FA pathway of DNA repair. The multiple lines of evidence linking RUNX proteins with the FA pathway of DNA repair are summarized here. 1. *Runx1*/*Runx3* (DKO) mice experienced mortality due to either BMF or a myeloproliferative disease, and cells derived from such mice had an elevated sensitivity to DNA ICLs, reminiscent of phenotypes manifested by FA patients. 2. RUNX proteins were poly(ADP) ribosylated or PARylated after DNA damage in a PARP-dependent manner. RUNX proteins interacted and promoted the recruitment of mono-ubiquitinated FANCD2 to sites of DNA ICLs and promoted efficient DNA repair. **3.** In the presence of DNA ICLs, RUNX3 has an increased interaction with the kinetochore complex (AURB, DSN1, CASC5, MIS12, ZWINT, PMF1, NSL1, and TPX2), the E2F7 transcription factor, and BLM (adapted from Tayet et al. [60]), of which the RUNX3-BLM interaction was shown to promote FANCD2 recruitment to sites of DNA ICLs. 4. RUNX1 walker domain mutations (G141 and R142 residues) from breast cancers were defective for DNA damage-dependent RUNX PARylation.

**Figure 6 cells-12-01106-f006:**
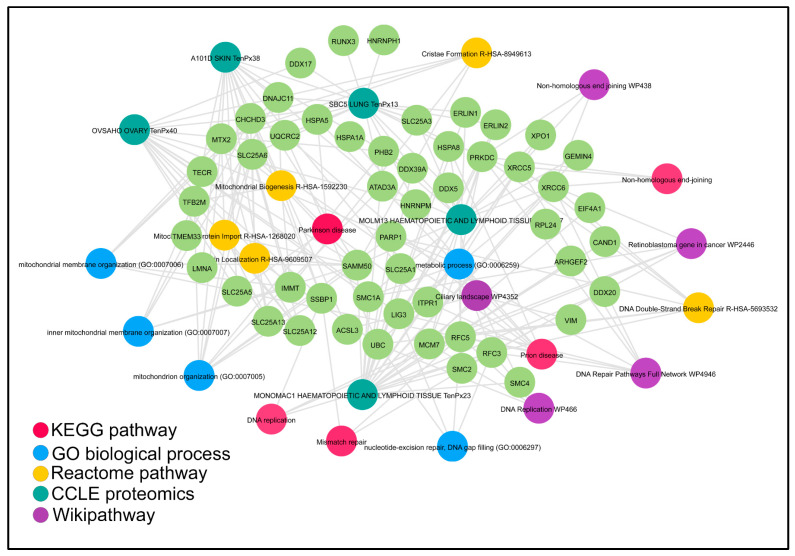
The interaction of RUNX3 with DNA replication and DNA repair factors in the absence of exogenous DNA damage. The SILAC mass spectrometry methodology was used to dissect the RUNX3 interactome in the absence of exogenous DNA damage. The list of RUNX3 interacting proteins from Tay et al., [60], Table S1, was analyzed using the EnrichR database [80]. Proteins indicated within green circles were retrieved as RUNX3-interacting proteins in co-immunoprecipitation studies. For the complete list of RUNX3-interacting proteins, please refer to [60].

**Figure 7 cells-12-01106-f007:**
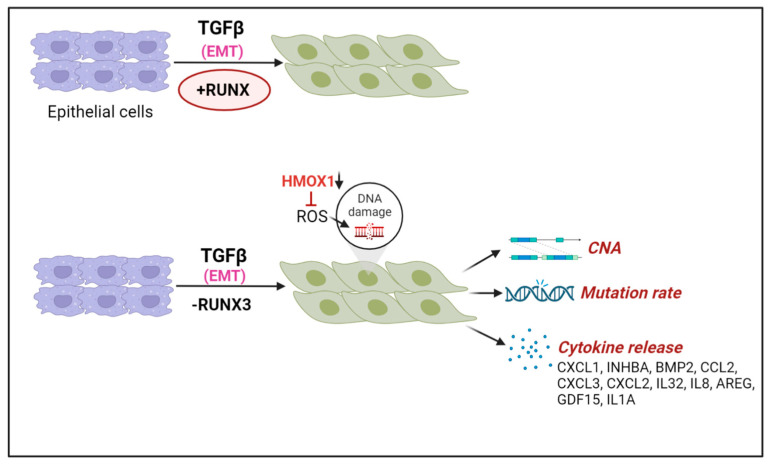
RUNX proteins guard genomic integrity after TGFβ exposure. The loss of RUNX proteins triggers dramatic DNA damage cells in lung epithelial cells undergoing TGFβ-dependent EMT. RUNX proteins reduce TGFβ induced-oxidative stress by the transcriptional induction of the antioxidant *HMOX1*. However, upon RUNX loss, TGFβ induced much higher levels of oxidative damage, DSBs, and acquisition of the senescence-associated secretory phenotype (SASP), represented by *CXCL1*, *INHBA*, *BMP2*, *CCL2*, *CXCL3*, *CXCL2*, *IL32*, *IL8*, *AREG*, *GDF15*, and *IL1A* upregulation. Since inflammation can induce DNA damage, *RUNX3* deficiency creates a self-re-enforcing feedback loop of inflammation and genomic instability.

**Table 1 cells-12-01106-t001:** A summary on the relationship between RUNX proteins, DNA repair, and cancer.

RUNX Models (Human Cancer/In Vivo/In Vitro)		Effects on DNA Repair Gene Expression/Genomic Integrity	References
RUNX1 fusion genes	RUNX1-ETO t(8;21)	DNA damage, DNA repair gene transcriptional impairment, mutator phenotype; ROS-associated SBS18 mutational signature; Impaired FANCD2 recruitment; PARPi sensitivity	[22,23,24,25,26,27,28,29,76]
	ETV6-RUNX1 t(12;21)	DNA repair gene transcriptional impairment and p53 pathway deregulation	[33,34]
	RUNX1-EVI1 t(3;21)	DNA repair gene transcriptional impairment (zebrafish model)	[38]
RUNX1 mutations	RUNX1-R174Q	DNA repair gene transcriptional impairment (iPSC model)	[46]
	RUNX1-C terminal deletions	DNA damage accumulation (murine stem cell model)	[45]
	RUNX1-R162K, R204Q, R107C	Mutational signature 9 in CML blast crisis patient samples	[47]
	RUNX1-G141, R142	Impaired FANCD2 recruitment expression of breast cancer walker domain mutations	[60]
RUNX3 mutation	RUNX3-R122C	Upregulation of MYC, a known activator of DNA replication stress (mouse model)	[51]
RUNX3 altered expression	RUNX3-transcriptional silencing	Copy number alterations and mutational accumulation (bladder, lung, liver, esophageal cancers)	[60]
	RUNX3 overexpression	Higher resistance to DNA damage-induced apoptosis in AML	[61]
RUNX2 upregulation	RUNX2 overexpression	Chemoresistance to adriamycin	[66]
Mouse model	*Runx1*/*Runx3* double knockout mice	Phenotypic resemblance to human Fanconi anemia	[76]
	*Runx2*-null osteoblasts	DNA damage accumulation and loss of telomeric integrity	[63]
Biochemical studies	RUNX interactome	Interaction of distinct DNA repair complexes in the presence and absence of DNA damage; RUNX1 interaction with Primpol	[60,88]
	RUNX interaction with p53	Regulates the transactivation of p53 target genes	[67,68,69,70,71,72]
Cytokine exposed cell model and human cancer	RUNX3 downregulation and cytokine exposure	Higher oxidative stress and DNA damage accumulation in RUNX3-deficient lung cancers exposed to TGF-beta	[89]

## Data Availability

Not applicable.

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
