# Peer review of "The RUNX Family of Proteins, DNA Repair, and Cancer"

_cells, 2023, doi:10.3390/cells12081106_

Round 1

Reviewer 1 Report

The author reviewed the RUNX family of proteins and their potential roles in genome stability and DNA repair. It is a well-written and well-organized manuscript. I recommend its acceptance with the following minor revisions.

Minor points:

·        A table summarizes all the RUNX1 and RUNX3 mutations and fusions in cancer

·        Fig 5 and Fig 6 are too blurry to read

Author Response

This author would like to thank the reviewer for the valuable and insightful comments. Please find below a point-by-point response to the reviewers.

Reviewer 1

The author reviewed the RUNX family of proteins and their potential roles in genome stability and DNA repair. It is a well-written and well-organized manuscript. I recommend its acceptance with the following minor revisions. 

Minor points:

  • A table summarizes all the RUNX1 and RUNX3 mutations and fusions in cancer

We agree that a table summarising all the RUNX1 and RUNX3 aberrations in cancer will be very useful. However, given the long list of such aberrations, they are probably beyond the scope of this review. Instead, a table that summarises the relationship between RUNX fusions, mutations and mouse models discussed in this review, have been summarised in Table 1.

  • Fig 5 and Fig 6 are too blurry to read

      Thank you for pointing this out. The figures have been replaced by high-resolution versions.

Reviewer 2 Report

Krishnan et. al provides a timely and well-written overview about the role of the RUNX proteins, including their roles in development and cancer, their potential function as either tumor suppressors or oncogenes, and their impact on DNA repair mechanisms. The review also highlights some of the specific pathways that RUNX proteins regulate to control the cellular response to DNA damage. The review is well organized covering most relevant aspects of the RUNX proteins. The figures provide an useful summary about the roles of the three RUNX proteins. 

Overall, the review would be a good addition to the literature.

I have following suggestions for improvement:

1) Clarify, why the RUNX proteins are presented in the order RUNX1, RUNX3 and RUNX2, and not RUNX1, RUNX2 and RUNX3. The author may have a valid reason for this order, so it would be helpful to understand the reasoning.

2) Include a mention of another alternative name for RUNX1-ETO, AML1-ETO, and incorporate some of the related papers that are not currently mentioned in the review, such as PMID: 27346355, 28360416

3) Provide more details and references about ETO as a transcriptional repressor. This will help readers understand this aspect of the RUNX1-ETO protein better.

3) Consider including a table that summarizes the most common translocations and mutations of the RUNX1 protein and their biological implications. (perhaps PMID: 19414342 can be used as a starting point.)

4) Remove Figure 5, as it is not very readable and the analysis is not very useful. The STRING database contains a lot of false positive interactions, and I would avoid using it.

5) The references to figures in lines 347, 366, 376, and 409 appear to be incorrect and should be corrected.

6) Line 434: Please correct E2F > E2F7

Author Response

Reviewer 2

Krishnan et. al provides a timely and well-written overview about the role of the RUNX proteins, including their roles in development and cancer, their potential function as either tumor suppressors or oncogenes, and their impact on DNA repair mechanisms. The review also highlights some of the specific pathways that RUNX proteins regulate to control the cellular response to DNA damage. The review is well organized covering most relevant aspects of the RUNX proteins. The figures provide an useful summary about the roles of the three RUNX proteins. 

 Overall, the review would be a good addition to the literature.

Thank you very much for the very useful comments and important additions requested to the revised version. Kindly find attached the point-point-response below.

I have following suggestions for improvement:

1) Clarify, why the RUNX proteins are presented in the order RUNX1, RUNX3 and RUNX2, and not RUNX1, RUNX2 and RUNX3. The author may have a valid reason for this order, so it would be helpful to understand the reasoning.

Thank you for raising this interesting point. From our review of the current literature, it seems like the roles of RUNX1 and RUNX3 are similar and many times even redundant in which one protein can replace the function of the other. On the other hand, the role of RUNX2 can be contrasting at times. For example, RUNX1 and RUNX3 upon association with p53 activate target gene transactivation, while RUNX2 prevents p53 transactivation. RUNX2 is mostly oncogenic while RUNX1 and RUNX3 are either tumor suppressors or oncogenic. Even though all three proteins share the RUNT domain and many other biochemical features, the divergence of their roles in a context-dependent manner remains unclear. Hence, we have organized the roles of these proteins as RUNX1, RUNX3 and RUNX2.

2) Include a mention of another alternative name for RUNX1-ETO, AML1-ETO, and incorporate some of the related papers that are not currently mentioned in the review, such as PMID: 27346355, 28360416. Provide more details and references about ETO as a transcriptional repressor. This will help readers understand this aspect of the RUNX1-ETO protein better.

Thank you for the important comment. A few sentences describing the RUNX1-ETO literature has been now included.

3) Consider including a table that summarizes the most common translocations and mutations of the RUNX1 protein and their biological implications. (perhaps PMID: 19414342 can be used as a starting point.)

We agree that a table summarizing all the RUNX1 and RUNX3 aberrations in cancer will be very useful. However, given the long list of such aberrations, they are probably beyond the scope of this review. Instead, a table that summarizes the relationship between RUNX fusions, mutations and mouse models discussed in this review, have been summarized in Table 1.

4) Remove Figure 5, as it is not very readable and the analysis is not very useful. The STRING database contains a lot of false positive interactions, and I would avoid using it.

Thank you for the comment. We agree that STRING based network analysis is not the best way to represent the RUNX3 co-immunoprecipitation dataset. However, it would be valuable to the reader if the RUNX3 interactome can be presented in the perspective of a interaction network. Thus, the EnrichR-KG database has been used to present a better analysis using four pathway analysis tools. 1. GO biological process 2. KEGG 3. Wikipathway 4. Reactome

5) The references to figures in lines 347, 366, 376, and 409 appear to be incorrect and should be corrected.

Thank you very much for pointing this out. These have been corrected in the revised version.

6) Line 434: Please correct E2F > E2F7

This typing error has been corrected in the revised version.

Reviewer 3 Report

In this manuscript, V. Krishnan reviews the involvement of RUNX proteins in cancer genomic instability. 

The manuscript is well written, easy to follow and well organized. Figures are well done and useful to summarize the main findings related to RUNX proteins.

Despite many reviews previously published on Cells journal and other journals covering overlapping topics are available in the literature, this manuscript details RUNX roles in genomic instability providing a comprehensive view.

Please check typos throughout the text (e.g. line 97, 242, 369, 385)

Figure 5 and 6 seem to be inverted

Author Response

Reviewer 3

In this manuscript, V. Krishnan reviews the involvement of RUNX proteins in cancer genomic instability. 

The manuscript is well written, easy to follow and well organized. Figures are well done and useful to summarize the main findings related to RUNX proteins.

Despite many reviews previously published on Cells journal and other journals covering overlapping topics are available in the literature, this manuscript details RUNX roles in genomic instability providing a comprehensive view.

Please check typos throughout the text (e.g. line 97, 242, 369, 385)

Figure 5 and 6 seem to be inverted

Thank you very much for the valuable comments. The manuscript has been proofread again and the figure numbers have been corrected.